# Chemotherapy for Gastric Cancer Is Not Solely the Domain of the Oncologist

**DOI:** 10.3390/cancers18010141

**Published:** 2025-12-31

**Authors:** Gabriel Samasca, Ioana Badiu Tisa, Calin Lazar, Ciprian N. Silaghi, Diana Deleanu, Adriana Muntean, Iulia Lupan

**Affiliations:** 1Department of Immunology, Iuliu Hatieganu University of Medicine and Pharmacy, 400006 Cluj-Napoca, Romania; gabriel.samasca@umfcluj.ro (G.S.);; 2Department of Nursing, Iuliu Hatieganu University of Medicine and Pharmacy, 400217 Cluj-Napoca, Romania; 3Department of Pediatrics 1, Iuliu Hatieganu University of Medicine and Pharmacy, 400370 Cluj-Napoca, Romania; 4Department of Medical Biochemistry, Iuliu Hatieganu University of Medicine and Pharmacy, 400349 Cluj-Napoca, Romania; 5Department of Internal Medicine, Regional Institute of Gastroenterology “Prof. Dr. Octavian Fodor”, 400162 Cluj-Napoca, Romania; 6Department of Molecular Biology, Babes-Bolyai University, 400084 Cluj-Napoca, Romania

**Keywords:** a multidisciplinary team, chemotherapy, gastric cancer

## Abstract

In the context of gastric cancer, chemotherapy is frequently employed as a systemic treatment, particularly when curative options are limited or not feasible. Patients diagnosed with cancer often seek to extend survival and may place considerable hope in chemotherapy as a key therapeutic avenue. The efficacy of chemotherapy depends on adequate recovery between treatment cycles, with the patient’s hematologic and systemic recovery status serving as key determinants of tolerance to subsequent doses and of overall treatment outcome. The administration of chemotherapy regimens frequently results in immunosuppression, and this compromised immune state elevates patients’ susceptibility to opportunistic infections, with implications for infection prevention and management. During intervals between chemotherapy cycles, access to rehydration infusions that provide vitamins B1, B6, and C is frequently limited, and payment for these adjunctive therapies is commonly borne by patients. Given the significance of the issue, urgent engagement by the medical community appears warranted; this article presents a potential solution.

## 1. Introduction

Cancers of the digestive tract, including gastric, colorectal, and hepatic malignancies, represent a substantial portion of global cancer morbidity and mortality, underscoring their public health significance. Although progress has been achieved in conventional therapies—chemotherapy, radiotherapy, surgical resection, and immunotherapy—clinical outcomes remain constrained by modest therapeutic efficacy, significant treatment-related toxicities, and a generally unfavorable prognosis. Systemic therapies for advanced gastric cancer (GC), comprising cytotoxic chemotherapy, molecularly targeted agents, and immunotherapies, have evolved substantially in recent years, expanding the therapeutic arsenal, improving efficacy and tolerability, and broadening options for patient management. The shift toward biomarker-guided and combination strategies has enhanced patient selection and enabled more durable responses, thereby reshaping the standard of care; ongoing translational and clinical research continues to refine optimal sequencing, integration with local therapies, and mechanisms of resistance, signaling a rapidly evolving and increasingly personalized treatment landscape for advanced GC [1]. Current oncologic guidelines designate the combination of immune checkpoint inhibitors (ICIs) and chemotherapy as the standard first-line (1L) therapy for advanced gastric or gastro-oesophageal junction (G/GEJ) cancer. Nevertheless, the observed efficacy is not optimal across the patient population, indicating a persistent need for strategies to enhance response rates and durability of benefit [2]. Conventional regimens of radiotherapy and chemotherapy are associated with substantial drawbacks that can limit therapeutic benefit and diminish patient well-being. Notable among these are pronounced toxicity with a broad spectrum of adverse effects, insufficient precision in targeting malignant tissue, and the potential emergence of drug resistance, collectively compromising tolerability, adherence, and quality of life (QOL), thereby underscoring the ongoing need for strategies that enhance targeting accuracy and minimize systemic toxicity [3].

## 2. Materials and Methods

Theme and objective.

This manuscript draws its theme from routine clinical practice. In GC, a substantial proportion of patients are discharged home between chemotherapy cycles to manage treatment-related adverse effects independently; in many cases, the medications prescribed to mitigate these toxicities fail to provide adequate relief. These adverse effects are well documented and affect an increasing number of patients. The study does not aim to identify novel aspects of care. Rather, its primary objective is to evaluate the adverse effects associated with chemotherapy in patients with GC. A secondary objective is to summarize current research in the field to contextualize the issue; however, this work is not intended to be a comprehensive review. Instead, the scope is restricted to examining interventions that reduce chemotherapy-associated toxicity in GC patients.

Methodology.

Our study employed a contemporary narrative review to identify studies published in 2025 that reported adverse events associated with chemotherapy for GC and were authored by clinical colleagues from countries other than our own. PubMed was searched using the term “side effects of chemotherapy in gastric cancer” to identify the most significant and recent publications on the topic. The inclusion criteria encompassed publications that collectively characterize the spectrum of chemotherapy-related toxicities and adverse reactions in GC and their implications for future treatment strategies and patient QQL. Eligible studies were screened for relevance, and the findings were synthesized to map the current knowledge base and identify gaps in the literature. Exclusion criteria were applied to all years except 2025. The analysis of all data retrieval from the PubMed database is presented in Figure 1 and Figure 2.

### 2.1. Is Helicobacter pylori Infection Eradicated?

*Helicobacter pylori* infection remains a major contributor to global morbidity, imposing a substantial health burden across diverse populations. The infection is highly prevalent, with the highest concentrations in low-and middle-income countries, and transmission often occurring in childhood, leading to decades-long carriage. Clinically, *H. pylori* is associated with gastritis and peptic ulcer disease and is linked to an elevated risk of non-cardia gastric adenocarcinoma and mucosa-associated lymphoid tissue (MALT) lymphoma, thereby driving symptomatic illness and cancer burden. Economically, both direct medical costs for diagnosis and treatment and indirect costs from lost productivity and longer-term complications contribute to a substantial economic impact [4]. Efforts to reduce this burden face significant regional variation in prevalence and asymptomatic carriage, compounded by rising antibiotic resistance that undermines eradication efforts and complicates standard regimens. This synthesis argues for comprehensive public health strategies that integrate improved diagnostics, locally adapted therapies, preventive approaches—including ongoing vaccination research—and enhanced surveillance with standardized reporting across regions. It also integrates evidence on eradication safety and effectiveness, emerging education modalities, and region-specific treatment considerations to inform policy and practice. The infection’s high prevalence in resource-limited settings, coupled with childhood transmission and prolonged carriage, underpins a persistent public health challenge. Understanding the interplay among epidemiology, clinical outcomes, therapeutic options, and health system factors is essential to reduce the burden and improve patient trajectories. *H. pylori* infection is widespread, with particularly high prevalence in resource-constrained settings. Transmission commonly occurs in childhood, leading to long-term carriage that sustains population-level disease risk. Clinically, the infection is linked to gastritis and peptic ulcer disease and is associated with an elevated risk of non-cardia gastric adenocarcinoma and MALT lymphoma. Collectively, these manifestations contribute to substantial symptomatic illness and cancer burden and drive significant direct and indirect costs to health systems and economies. The economic impact encompasses direct costs for diagnosis, treatment, and follow-up, as well as indirect costs from lost productivity and the long-term sequelae of gastroduodenal diseases and cancer; accurate quantification requires robust surveillance and standardized reporting across regions. Public health strategies to reduce the burden are challenged by regional variation, asymptomatic carriage, and rising antimicrobial resistance, which undermine eradication success and complicate regimens. Priorities include improved diagnostic methods for timely and accurate detection, evidence-based, locally adapted treatment protocols that account for regional resistance patterns, preventive approaches including ongoing vaccination research, and enhanced surveillance with standardized reporting to monitor disease burden and intervention impact over time [5,6,7]. Eradication of *H. pylori* has been shown to reduce GC risk, though the extent of risk reduction varies by setting. Randomized and observational studies generally support a protective association, yet heterogeneity in study populations, baseline mucosal status, adherence to therapy, and follow-up duration complicate causal interpretation. While effective antimicrobial regimens exist, eradication rates remain suboptimal largely due to imperfect patient adherence.

Emerging modalities of patient education, particularly new media-based education (NME), hold promise for improving understanding, engagement, and adherence to treatment regimens. Implementing NME in clinical settings may facilitate patient-centered communication, enable tailored information delivery, and promote sustained adherence, with potential improvements in therapeutic outcomes; personalized, interactive platforms (e.g., WeChat-based interventions) show substantial potential for enhancing patient engagement and treatment outcomes [4]. Proton pump inhibitors (PPIs) are among the most widely prescribed medications globally due to their efficacy in symptom control and mucosal healing across acid-related disorders, including gastro-oesophageal reflux disease, peptic ulcer disease, *H. pylori* eradication therapy, functional dyspepsia, and gastroprotection in high-risk patients. However, extended use beyond approved indications has raised safety concerns. Observational data link chronic PPI use with adverse outcomes such as enteric infections (notably *Clostridioides difficile*), micronutrient deficiencies (e.g., hypomagnesemia, vitamin B12 deficiency), osteoporotic fractures, chronic kidney disease, dementia, and potential associations with gastric and colorectal cancer. Causality remains uncertain due to observational design, residual confounding, and indication bias; nonetheless, risk–benefit considerations in long-term therapy warrant cautious assessment [5]. Regimen selection continues to evolve, with dual therapy using vonoprazan (Vo) and amoxicillin (Amx) showing promising efficacy for *H. pylori* eradication, though regional validation is needed in high-burden settings. The extent to which Vo-Amx plus bismuth (Bis) regimens enhance eradication remains to be clarified; in some regions, Vo-containing regimens have demonstrated non-inferiority to bismuth quadruple therapy (BQT) and could reduce antibiotic exposure by one agent per patient relative to BQT [6]. Traditional triple therapy (CTT) has suffered reduced effectiveness in settings with higher antimicrobial resistance, while quadruple therapy, sequential therapy, and high-dose PPI regimens have yielded superior outcomes compared with standard-dose therapies in randomized trials and comparative studies, with faster symptom resolution and lower relapse rates. These advantages are attributed to stronger and more sustained acid suppression, improved pharmacodynamic activity, and enhanced mucosal healing; however, these regimens may entail greater tolerability concerns, higher adverse event rates, increased costs, and adherence challenges. The choice of regimen should be informed by a careful, context-specific risk–benefit assessment and aligned with established guidelines [8]. Region-specific treatment strategies guided by antimicrobial susceptibility testing are recommended to optimize clinical outcomes and minimize the emergence of resistance, thereby supporting more effective eradication while preserving antibiotic effectiveness for future use.

### 2.2. Adverse Effects of Drugs

#### 2.2.1. Anti-HER-2 Drugs

This synthesis collates recent clinical and preclinical evidence on targeted therapies in G/GEJ cancers, with a focus on antibody-drug conjugates (ADCs), their counterparts radioimmunoconjugates, and evolving combination strategies involving human epidermal growth factor receptor 2 (HER2) and Claudin-18.2 (CLDN18.2) targets. Across these data, ADC-based approaches often show promising antitumor activity and manageable safety profiles, particularly when combined with immune checkpoint blockade, and when sequencing strategies favor ADC initiation. In HER2-overexpressing, pre-treated advanced G/GEJ, RC48 (an anti-HER2ADC) combined with a programmed cell death protein 1 (PD-1) inhibitor demonstrated superior efficacy compared with RC48 monotherapy. Specifically, the combination achieved an objective response rate (ORR) of 41.7% versus 9.5% with RC48 alone (*p* = 0.011). Disease control rate (DCR) did not differ significantly between groups (75.0% vs. 57.1%; *p* = 0.162). Median overall survival (OS) was longer with the combination (13.2 months vs. 7.1 months; *p* = 0.040), and median progression-free survival (PFS) trended in favor of the combination (5.8 vs. 2.9 months; *p* = 0.142). Subgroup analyses within the combination cohort suggested that patients with initial HER2 immunohistochemistry (IHC) 3+, those receiving second-line therapy, or with liver metastases tended to experience more favorable outcomes. Safety profiles favored RC48 plus PD-1 inhibition relative to RC48 alone, indicating that this combination is tolerable in this setting [9]. In the CLDN18.2 space, preclinical and translational data indicate that ADCs targeting CLDN18.2 may confer superior antitumor activity and a more favorable safety profile compared with radionuclide-drug conjugates (RDC) derived from the same CLDN18.2-directed monoclonal antibody. Moreover, sequential therapy initiating with an ADC appears more advantageous than strategies starting with an RDC. Although the ADC → RDC sequence did not significantly outperform ADC monotherapy in the evaluated model, it may still represent a viable subsequent treatment option, and these findings support prioritizing ADC-based approaches in CLDN18.2-directed therapy and exploring ADC-initiated sequences in future work [9]. The broader HER2 landscape in metastatic disease underscores ongoing challenges and opportunities. Overexpression and/or amplification of HER2 remains a key oncogenic driver in metastatic breast cancer and in metastatic GEJ/esophageal adenocarcinomas, driving a continuum of therapies including HER2-directed monoclonal antibodies, ADCs, and tyrosine kinase inhibitors (TKIs). Emerging agents such as Zongertinib—a novel, irreversible, HER2-selective TKI that spares Epidermal Growth Factor Receptor (EGFR) signaling—aim to extend the therapeutic window by reducing EGFR-related toxicities [10]. In HER2-negative disease, combinations such as surufatinib with paclitaxel for second-line therapy in unresectable or metastatic gastric/GEJ adenocarcinoma yielded an ORR of 25.0% (95% CI: 11.5–43.4%), DCR of 87.5% (95% CI: 71.0–96.5%), median PFS of 5.7 months (95% CI: 4.7–6.9), and median OS of 10.8 months (95% CI: 7.0–17.2). Notably, among patients with prior immunotherapy exposure (*n* = 26), median OS extended to 14.4 months (95% CI: 8.5–not estimable). Treatment-related grade ≥3 adverse events occurred in 54.3% of patients, with neutropenia (40.0%), leukopenia (34.3%), and hypertension (11.4%) being most common, underscoring a manageable safety profile that supports further investigation in randomized trials [11]. Trastuzumab deruxtecan (T-DXd) has been evaluated as second-line therapy in Western patients with HER2-positive unresectable or metastatic gastric/GEJ adenocarcinoma progressed after trastuzumab-based 1Ltherapy. In the DESTINY-Gastric02 (DG-02) study, efficacy was contextualized against a historical comparator of ramucirumab plus paclitaxel (Ram + Pac). In this context, T-DXd demonstrated improved OS relative to Ram + Pac, supporting its role as a second-line option in this setting [12]. The incremental value of dual HER2 blockade—pertuzumab plus trastuzumab added to chemotherapy (P + T + cT)—has been examined across a pooled analysis of four trials (*n* ≈ 1225). The combination yielded a significant survival benefit, with an OS hazard ratio (HR) of 0.77 (95% CI 0.69–0.86) and improved PFS HR 0.73 (95% CI 0.62–0.85). Pathological response rates favored dual blockade OR 1.62 (95% CI 0.98–2.66). However, the addition of dual HER2 blockade increased the risk of any-grade adverse events (OR 1.48, 95% CI 1.32–1.66), with higher incidences of diarrhea, hypokalemia, fatigue, and pulmonary infection. Importantly, rates of serious adverse events did not differ significantly between groups OR 0.87 (95% CI 0.51–1.49). Collectively, these findings indicate that dual HER2 inhibition combined with chemotherapy can improve survival and pathological response in HER2-positive gastric/GEJ cancers, albeit at the cost of increased non-serious toxicities [13]. Beyond targeted agents, interactions between anti-angiogenic therapy and other medications warrant consideration. Although primary pharmacodynamic effects of most anticancer drugs are well characterized, concomitant medications used to mitigate adverse effects can attenuate efficacy. In particular, anti-VEGF therapies may have altered outcomes when administered with gastric acid secretion inhibitors, with evidence suggesting involvement of estrogen receptor signaling in this interaction [14]. High HER3 expression across various tumor types has been linked to adverse clinical outcomes, and, to date, no HER3-directed antibody-drug conjugate has been approved. A HER3-directed ADC (HER3-DXd) comprises an anti-HER3 mAb linked to a topoisomerase I inhibitor payload via a stable tetrapeptide-based cleavable linker and remains investigational [15].

#### 2.2.2. Chemotherapeutic Agents

This synthesis surveys the current landscape of chemotherapeutic strategies for GC, highlighting efficacy and toxicity concerns of established agents, and examining emerging adjuvant and targeted delivery approaches that may enhance therapeutic outcomes while mitigating adverse effects. Among widely used drugs, doxorubicin is employed across multiple malignancies, including breast, lung, gastric, ovarian, and thyroid cancers, as well as lymphoma; however, its clinical utility is constrained by significant toxicities such as cardiotoxicity, bone marrow suppression, and nephrotoxicity, which together narrow the therapeutic window and drive the pursuit of protective strategies and dosing paradigms [16]. Oxaliplatin, a platinum-based agent, is applied in colorectal and GCs; preclinical and clinical data suggest that oxaliplatin-containing regimens can achieve enhanced tumor suppression, reflected by notable reductions in proliferation indices such as Kiel 67 (Ki-67) and mitotic activity, although exposure to chemotherapy remains a challenge due to adverse events that frequently lead to treatment discontinuation [17]. Cisplatin (DDP) continues to be a foundational chemotherapeutic for GC; nonetheless, its clinical use is hampered by toxicity and the development of chemoresistance, which limit its long-term effectiveness. In parallel, natural product-derived compounds are being explored as adjuvants; psoralidin (PSO), the principal bioactive constituent of Psoralea corylifolia, has demonstrated antitumor activity, and emerging evidence indicates that combining PSO with DDP yields synergistic antitumor effects in GC cells via long-chain-fatty-acid—CoA ligase 4 (ACSL4)-dependent ferroptosis. These findings propose PSO as a potential non-toxic adjuvant to augment DDP efficacy and possibly mitigate DDP-associated toxicity, though further mechanistic investigations and in vivo validation are required to establish translational potential and to determine optimal dosing strategies [18]. To address systemic toxicity associated with hydroxycamptothecin (HCPT), a targeted drug-delivery system was developed in which hollow mesoporous polydopamine (HMPDA) surfaces were functionalized with an Angiopoietin-2 (ANGPT2)-specific peptide (GSF; GSFIHSVPRH) and loaded with HCPT and indocyanine green (ICG), yielding the ICG-GSF-HMPDA@HCPT nanoplatform. Comprehensive characterization confirmed successful surface modification and co-loading of payloads, while blood biochemical assays indicated favorable biosafety and acceptable systemic tolerance. Collectively, HCPT-loaded HMPDA functionalized with an ANGPT2-specific peptide represents a promising nanoscale carrier for targeted GC therapy, enabling precise delivery and theranostic capabilities through ICG. Future investigations should assess in vivo efficacy, pharmacokinetics, and long-term safety to solidify the translational potential of this delivery system [19].

#### 2.2.3. Neoadjuvant Chemotherapy

GC remains a highly prevalent malignancy worldwide, with locally advanced gastric cancer (LAGC) carrying a particularly poor prognosis largely because achieving an R0 resection is frequently unattainable. Neoadjuvant chemotherapy (NAC) can improve survival, yet its effectiveness is limited, underscoring the need for more effective multimodal strategies. ICIs have demonstrated activity in advanced GC, but their role in neoadjuvant therapy (NAT) for LAGC remains unsettled. Emerging evidence suggests that NAT regimens incorporating ICIs can substantially improve pathologic complete response (pCR) rates and R0 resection rates in LAGC without increasing perioperative risk; however, these advantages have not consistently translated into improvements in short-term survival [20]. The completion rate of adjuvant chemotherapy for GC remains suboptimal, a problem that is particularly pronounced in elderly patients. Although NAC for locally advanced GC has shown promise, data in older populations are limited. In a retrospective analysis of 38 clinically staged II/III GC patients who received NAC, participants were stratified into non-elderly (<75 years, *n* = 27) and elderly (≥75 years, *n* = 11) groups. The elderly cohort exhibited poorer ECOG performance status (*p* = 0.016). While all non-elderly patients completed ≤3 NAC cycles, a greater proportion of elderly patients received 4 cycles (*p* = 0.0047). Per-cycle dose intensities of S-1 (*p* = 0.0003) and oxaliplatin (*p* = 0.0018) were significantly lower in the elderly group. Adverse events and treatment efficacy were comparable between groups. In this NAC cohort for cStage II/III GC, advanced age was associated with higher total cycle numbers and reduced per-cycle dosing, without detectable differences in safety or early efficacy. Prospective studies are warranted to optimize NAC regimens for elderly patients [21]. Neoadjuvant immunochemotherapy may confer clinically meaningful benefits for LAGC, as suggested by signals of improved pCR rates and greater tumor downstaging (T downstaging), coupled with an acceptable safety profile. These preliminary findings support the potential role of this approach within multimodal management strategies for LAGC; however, confirmation in prospective, randomized trials are needed to establish definitive efficacy, optimize regimens, and define appropriate patient selection criteria [22]. Across NAT for LAGC, incorporating chemotherapy, molecular targeted therapy, and immunotherapy has demonstrated substantial improvements in radical (R0) resection rates and prognosis. Compared with NAC alone, the combined use of targeted therapy and/or immunotherapy yields greater tumor downstaging and higher pCR rates, although vigilance for severe treatment-related adverse events remains essential [23].

Nivolumab, a human anti-PD-1 monoclonal antibody, reinvigorates T-cell-mediated antitumor responses and has substantially transformed the cancer immunotherapy landscape. Despite broad clinical benefits across multiple malignancies, nivolumab can cause immune-related adverse events (irAEs), with immune-mediated hepatitis (IMH) being particularly notable due to potential progression to acute liver injury or life-threatening deterioration [24]. Identifying patients who derive benefit from anti-PD-1/ Programmed Death-Ligand 1 (PD-L1) therapies remains challenging. One study evaluated nivolumab monotherapy in patients with a high PD-1–positive CD8/Treg ratio in advanced non-small cell lung cancer (NSCLC) and GC, reporting treatment-related AEs in 13 of 19 patients overall (68.4%), including 5 of 6 NSCLC patients and 8 of 13 GC patients [25]. In advanced GC treated with a combination of chemotherapy and nivolumab, the appearance of irAEs has been associated with improved survival outcomes [26]. Several barriers limit the universal benefit of adjuvant chemotherapy. Some cancer types—such as renal cell carcinoma, hepatocellular carcinoma, and cholangiocarcinoma—lack highly effective regimens, while patient- and treatment-related factors, including advanced age, poor performance status, and substantial adverse reactions, can preclude complete chemotherapy. Although targeted therapies and PD-1/PD-L1 inhibitors have broadened therapeutic options, their efficacy frequently depends on molecular biomarkers and genetic testing, and their benefits can be constrained by drug resistance and immune-related toxicity. In contrast, cytokine-induced killer (CIK) cell therapy, which involves reinfusion of highly activated CD3+ T cells, has demonstrated a favorable safety profile with an absence of severe side effects in the contexts described [27]. Therapeutic challenges also arise with immune checkpoint inhibitor–induced autoimmune hematologic toxicities, such as immune checkpoint inhibitor–induced autoimmune hemolytic anemia (ICI-AIHA) in elderly patients. Immunosenescence and diminished hematopoietic reserve complicate risk stratification and management, highlighting the urgent need for mechanistic investigations, development of individualized immunotherapeutic strategies that preserve oncologic efficacy while reducing hematologic toxicity, and vigilant hematologic monitoring in vulnerable populations [28].

Cytokine release syndrome (CRS), a hyperinflammatory condition most commonly observed after chimeric antigen receptor (CAR-T) cell therapy in hematologic malignancies, has also been observed in contexts suggesting immune-mediated etiologies. Clinicians should maintain a high index of suspicion for CRS when GC patients present with fever and systemic inflammation after immunotherapy, as timely recognition and differentiation from infection are critical for personalized management [29]. ICIs can also cause adrenal insufficiency (AI), a rare but potentially life-threatening irAE. AI often presents with nonspecific symptoms, making prompt endocrine evaluation essential, particularly when multiple symptoms co-occur in patients receiving ICI therapy. The adoption of standardized diagnostic pathways and robust interdisciplinary communication is important to improve recognition and clinical outcomes for this serious irAE [30].

Metronomic chemotherapy has shown potential to enhance the efficacy of PD-1–targeted agents but has not yet been studied in gastrointestinal (GI) cancers. A recent investigation examined the feasibility of metronomic capecitabine plus camrelizumab as salvage therapy for late-stage GI cancer and explored the roles of body composition and lipid metabolism in treatment outcomes. In eligible GI cancer patients with disease progression after standard chemotherapy, the regimen demonstrated tolerability and encouraging efficacy, with a 19.2% (ORR; 5 of 26) and five grade ≥3 treatment-emergent adverse events (TEAEs) among 26 patients. These findings suggest feasibility and potential benefit, warranting further study into how body composition and lipid metabolism influence outcomes [31].

#### 2.2.4. Targeted Therapy

Imatinib therapy for gastrointestinal stromal tumors (GIST) is associated with substantial toxicities, including myelosuppression, edema, and hypersensitivity, with notable interpatient variability. In a retrospective cohort of 154 patients with GIST treated with imatinib, significant associations were observed between myelosuppression and imatinib plasma concentration, as well as associations with genetic polymorphisms in Fms-related tyrosine kinase 1 (FLT1), mitogen-activated protein kinase 1 (MAPK1), the platelet-derived growth factor receptor beta (PDGFRB), and transforming protein 1 (SHC1). Peripheral edema was more prevalent in female patients and was linked to PDGFRB polymorphisms, while hypersensitivity correlated with polymorphisms in EGFR and Chemokine (C-X-C motif) ligand 14 (CXCL14) [32]. Succinate dehydrogenase (SDH)-deficient GISTs are generally resistant to targeted therapies with TKIs such as imatinib, and no standard therapeutic options exist for advanced SDH-deficient GISTs. The precise oncogenic mechanisms by which SDH mutations drive GIST pathogenesis remain incompletely understood. Olverembatinib, a novel multikinase inhibitor, has demonstrated encouraging activity in the setting of imatinib-resistant GIST. In this context, olverembatinib was generally well tolerated; TEAEs (occurring in ≥20%) included elevations in hepatic transaminases, leukocytosis and neutrophilia, anemia, and pyrexia. Among SDH-deficient GISTs, confirmed partial responses were observed in 6 of 26 evaluable patients (ORR, 23.1%; 95% CI, 9.0–43.7%), with an additional 16 patients (61.5%) not progressing during the first 6 months of treatment. This yielded a clinical benefit rate of 84.6% (95% CI, 65.1–95.6%), and the mPFS was 25.7 months (95% CI, 12.9–NR) [33].

#### 2.2.5. Other Drugs

Incretins are a class of metabolic hormones that decrease blood glucose levels. This synthesis examines the beneficial and detrimental effects of incretins in cancer development and therapy. Some studies have suggested an increased risk of pancreatic, thyroid, cholangiocarcinoma, and colorectal cancers associated with incretin therapies, particularly in individuals with genetic predispositions. Conversely, other research has demonstrated anticancer effects in prostate, breast, ovarian, and other cancers through mechanisms such as inhibition of tumor cell proliferation, induction of apoptosis, and modulation of immune responses. Although incretin-based drugs may pose tissue-specific cancer risks in susceptible individuals, their therapeutic potential in mitigating other cancer types appears promising. Overall, the evidence suggests that the benefits of incretins may outweigh the risks when patient-specific factors are carefully considered [34].

### 2.3. Adverse Effects of Surgery

This text presents an integrated synthesis of several areas within oncologic and GI surgery research, focusing on perioperative chemotherapy, postoperative complications, and novel therapeutic approaches. To evaluate the toxicity profile and clinical outcomes of perioperative FLOT (fluorouracil, leucovorin, oxaliplatin, and docetaxel) chemotherapy in elderly patients (≥65 years) with locally advanced gastric adenocarcinoma, a comparative analysis was conducted between patients aged ≥65 and those <65 who received neoadjuvant FLOT followed by surgery and adjuvant therapy. Endpoints included the mean number of FLOT cycles, cumulative chemotherapy exposure, and mOS. Statistical significance was assessed using conventional *p*-values. Results indicated that patients aged ≥65 received fewer FLOT cycles than those <65 (mean 4.5 vs. 7 cycles, respectively; *p* = 0.03). Despite the lower treatment intensity and reduced cumulative dose in the elderly group, there was no difference in median OS between the two age cohorts (*p* = 0.68). In this cohort, perioperative FLOT with fewer cycles in elderly patients did not compromise clinical outcomes, as evidenced by comparable median OS to younger patients [35]. A separate retrospective evaluation of Turkish (Tp) and German (Gp) patients with resectable GC or GEJ who received perioperative FLOT found that ethnicity demonstrated limited prognostic impact; however, lymph node status remained the strongest determinant of outcomes in perioperatively treated patients with GC/GEJ. Intensified systemic therapy did not improve the rate of major pathological response (mPR) and was associated with worse OS, while achieving mPR was not independently linked to OS. Pathological nodal positivity and signet-ring cell carcinoma emerged as key predictors of poor survival. Collectively, these findings underscore the importance of nodal risk stratification and tailored follow-up over therapy intensification in guiding management for resectable GC/GEJ [36].

The perioperative use of ICIs in GC has introduced substantial therapeutic challenges. Given the marked heterogeneity of GC, there is an urgent need to identify patients most likely to benefit from ICIs with precision through TME profiling, molecular biomarker screening, and clinically oriented subgroup analyses. Concurrently, the potential benefits of ICIs in locally advanced GC/GEJ must be weighed carefully against their adverse effects and the considerable financial toxicity associated with their use [37]. Several perioperative factors have been identified as determinants of postoperative outcomes. Advanced age, elevated body mass index (BMI), and delayed postoperative ambulation (>48 h) have been recognized as risk factors for Roux-en-Y stasis syndrome (RSS), whereas higher serum albumin levels, tumors located in the gastric body, laparoscopic surgery, the use of a linear stapler, and fixation of the duodenal stump have been associated with a reduced risk of RSS, representing protective factors [38]. Small intestinal bacterial overgrowth (SIBO) is recognized as a potential complication following upper GI surgery. The reported prevalence is approximately 31%, with the highest incidence (43%) observed in patients who underwent metabolic surgery. Adjuvant radiotherapy or chemotherapy has been associated with an increased risk of SIBO, and extensive small bowel resection or exclusion has been strongly linked to heightened risk. The existing diagnostic modalities for SIBO exhibit limited sensitivity and specificity, underscoring the need for early screening and the standardization of diagnostic techniques to improve patient management and outcomes [39].

Advances in radiotherapy adjuvant strategies include the combination of radiosensitizing carbon monoxide (CO) and radioprotective hydrogen (H_2_) for whole-abdominal radiotherapy in the treatment of peritoneal metastases from colon cancer. A novel dual-gas nanosystem has been developed in which unstable iron carbonyl (FeCO) and ammonium borane (AB) are co-encapsulated within a stable silica matrix to enable on-demand release of CO and H_2_. Designed to prevent premature gas release in acidic gastric environments and to sustain H_2_ release for more than 50 h in the intestinal tract, this system aims to provide protection against radiotherapy-induced intestinal injury. Upon X-ray irradiation at the tumor site, the system rapidly releases CO to augment radiosensitivity while concurrently releasing H_2_ to promote repair of radiation-induced intestinal damage. Collectively, these features suggest that the dual-gas nanosystem may enhance the therapeutic index of abdominal radiotherapy by simultaneously increasing tumor radiosensitivity and mitigating normal-tissue toxicity [40]. Regarding intraperitoneal chemotherapy (IPC), including the hyperthermic variant (HIPEC), the available evidence is characterized as very low certainty. Prophylactic and therapeutic IPC may be associated with improved survival but may have little to no effect on anastomotic leakage. Prophylactic IPC appears to have little to no effect on tumor recurrence and intra-abdominal abscess. Therapeutic IPC similarly shows little to no effect on serious adverse events, although limited evidence suggests that IPC may delay tumor progression and have little to no impact on health-related QOL [41].

### 2.4. Adverse Effects of Superselective Transarterial Chemoembolization

In a cohort of 16 patients who underwent superselective transarterial chemoembolization (SSTACE), safety outcomes indicated that all SSTACE-related adverse events were manageable. The most frequently reported adverse events were nausea (7/16, 43.8%), vomiting (6/16, 37.5%), and abdominal pain (6/16, 37.5%). No hepatic dysfunction was observed. Among the surgical patients in the cohort, there were no major postoperative complications. These findings suggest that SSTACE could potentially offer a novel therapeutic approach for patients with locally advanced unresectable GC or GEJ cancer who have shown limited response to systemic chemotherapy or immunotherapy, characterized by stable disease or progressive disease [42].

### 2.5. Other Adverse Events

Anemia and skeletal complications, along with treatment-related ocular and systemic toxicities, represent significant challenges for patients with GI cancers undergoing chemotherapy. This synthesis reviews evidence on iron deficiency anemia management with ferric carboxymaltose (FCM), cancer-associated osteoporosis risk and bone mineral density (BMD) changes, rare ocular adverse events linked to oxaliplatin-based regimens, and factors contributing to cancer-related fatigue (CRF). Together, these findings underscore the need for comprehensive, multidisciplinary strategies to optimize hematologic status, bone health, vision, and QOL in this patient population.

Anemia remains a substantial impediment to QOL and the continuity of cancer treatment in patients with GI malignancies undergoing chemotherapy. Conventional therapeutic options—red blood cell transfusions, erythropoiesis-stimulating agents, and oral iron—are constrained by thrombotic and other risks, supply limitations, and tolerability issues. FCM has been evaluated as a targeted intervention for iron deficiency anemia in patients with unresectable or recurrent gastric or colorectal cancer receiving chemotherapy, with the objective of improving hematologic status and patient well-being. In this protocol, participants were to receive a total dose of 1500 mg FCM administered as three 500 mg infusions, each given at least seven days apart and completed within 29 days from Day 1. The primary endpoint was the rate of hemoglobin (Hb) improvement at 8 weeks. The study design incorporated a sample size of 50 patients to detect a 33% improvement rate versus a 15% threshold with 80% power. Secondary endpoints included Hb improvement rates at 4 and 12 weeks, and QOL assessments at 4, 8, and 12 weeks. These elements reflect a rigorous approach to quantify hematologic and patient-centered outcomes in a population at high risk for persistent anemia during chemotherapy [43]. To quantify the association between cancer status and osteoporosis risk, as well as BMD changes, in comparison with non-cancer controls, data indicate a substantially elevated risk of osteoporosis among cancer patients. Cancer was associated with a 6.8-fold increase in osteoporosis risk (*p* < 0.001; 95% CI: 4.024–11.494). BMI appeared to exert a protective effect, with each unit increase in BMI associated with a 4.5% reduction in the odds of osteoporosis (OR 0.955; 95% CI: 0.915–0.997; *p* = 0.036). Lumbar osteoporosis and osteopenia were 9.1-fold and 2.3-fold more prevalent in cancer patients, respectively. Moreover, lumbar and femoral BMD were reduced by 26.2% and 9.2%, respectively, compared with controls. These findings underscore the critical importance of proactive bone health assessment and intervention in oncology care, given the upsurge in fracture risk and related morbidity among cancer survivors and those undergoing ongoing treatment. The observed associations suggest that routine screening for osteoporosis and BMD monitoring should be integrated into oncologic care, particularly for patients with GI cancers receiving cytotoxic chemotherapy. Interventions may include nutritional optimization, vitamin D and calcium repletion where appropriate, physical activity programs tailored to individual capabilities, and judicious consideration of pharmacologic anti-osteoporotic therapies in high-risk individuals. Addressing bone health is essential not only for reducing fracture risk but also for maintaining overall QOL and functional independence in cancer patients [44]. Ocular toxicities linked to systemic chemotherapy, though relatively rare, are increasingly recognized in clinical practice and warrant heightened vigilance. A notable case illustrates severe ocular toxicity associated with oxaliplatin-based chemotherapy, manifesting as multiple iris cysts, anterior segment inflammation, exudative retinal detachment, and lens opacity in a 47-year-old woman with adenocarcinoma of the esophagogastric junction who underwent laparoscopic radical gastrectomy and subsequent adjuvant oxaliplatin-based chemotherapy. Following three cycles, she experienced acute bilateral vision loss. Laboratory investigations showed markedly elevated systemic and aqueous humor interleukin-6 (IL-6) levels, while orbital imaging ruled out ocular metastasis. This case emphasizes the potential for oxaliplatin-related ocular toxicity and the need for prompt chemotherapy modification and timely ophthalmologic evaluation to avert irreversible visual impairment. Clinicians should maintain a high index of suspicion for ocular symptoms in patients receiving oxaliplatin-based regimens and pursue early ophthalmologic consultation when vision changes arise [45]. Multivariate analyses in GC patients undergoing chemotherapy identified several independent factors associated with higher CRF. Female sex, nutritional risk status, poor sleep quality, depressive symptoms, and pain were each independently linked to greater CRF. CRF is highly prevalent in this patient population and arises from a complex interplay of nutritional status, sleep, mood, and pain. These findings support the need for early identification of high-risk individuals and the development of personalized, targeted interventions to mitigate CRF. Integrating nutritional support, sleep hygiene strategies, psychosocial care, and effective pain management into oncologic care may help reduce CRF and improve overall functional outcomes [46].

## 3. Future Research Directions for Reducing Chemotherapy Side Effects

Cellular senescence is characterized by a predominantly irreversible arrest of the cell cycle and exerts a dual influence on cancer progression through the senescence-associated secretory phenotype (SASP). The SASP comprises an extensive repertoire of bioactive mediators—including cytokines, chemokines, growth factors, and proteases—that can markedly affect the tumor microenvironment (TME) by modulating inflammatory signaling, immune surveillance, extracellular matrix remodeling, and angiogenesis, thereby contributing to both tumor suppression and tumor promotion depending on cellular and contextual factors. Although SASP initially contributes to tumor suppression by recruiting immune effector cells and inhibiting cancer cell proliferation, its prolonged presence in the TME can promote tumor growth, metastasis, and resistance to therapy. Therapy-induced senescence is a frequently observed consequence of oncologic treatment and may contribute to an expanded population of senescent cells, accompanied by a pro-tumorigenic SASP. Recent research indicates that targeting the SASP may improve cancer therapy; by modulating SASP factors, such interventions could attenuate pro-tumorigenic signaling and enhance the responsiveness of tumors to conventional treatments, thereby offering a promising avenue for improving patient outcomes. Among therapeutic strategies under investigation, senolytic therapies selectively eradicate senescent cells, whereas senomorphic agents attenuate the SASP without inducing cytotoxicity. Additionally, combination approaches targeting SASP have been explored for oncotherapeutic applications [47].

Wedelolide A (WA), a sesquiterpene lactone isolated from *Sphagneticola trilobata*, has been investigated for its antitumor effects. WA exhibits potent antitumor activity by inducing both apoptosis and ferroptosis through mechanisms involving oxidative stress and mitochondrial dysfunction, with involvement of the Keap1/Nrf2/HO-1 signaling axis, highlighting its potential as a therapeutic candidate for GC therapy [48]. In Japan, Kampo medicine (Japanese herbal medicine) is frequently used to manage cancer-treatment–related side effects and to improve QOL; however, nationwide trends in Kampo use among inpatients with cancer remain poorly understood. This analysis shows that Kampo medicines were prescribed in 13.6% of hospitalizations, were more common among older adults and patients with colorectal cancer, and the overall prescription proportion increased from 2010 to 2017 before plateauing. The five most prescribed Kampo medicines were Dai-ken-chu-to, Gosha-jinki-gan, Rikkunshi-to, Shakuyaku-kanzo-to, and Hange-shashin-to. Prescription patterns varied by age group, cancer type, cancer stage, disease status, and cause of hospitalization. Dai-ken-chu-to use shifted from postoperative to chronic care, while Gosha-jinki-gan prescriptions gradually declined. Over the past fourteen years, Kampo prescription patterns among inpatients with cancer have changed, reflecting changes in patient demographics and treatment strategies, and suggesting that Kampo medicines are selectively used as supportive care tailored to specific clinical situations. These findings highlight the evolving role of traditional medicine in modern cancer care in Japan [49]. Self-management is essential for cancer patients undergoing chemotherapy to mitigate treatment-related adverse effects and enhance QOL. This study evaluated a smartphone-based mobile application designed to support the self-management of chemotherapy side effects in GC patients, leveraging the widespread use and capabilities of modern smartphones. A user-centered evaluation assessed the app’s acceptability and functionality, with satisfaction regarding its features and capabilities reported at 98%, indicating a high level of acceptance among users. The findings suggest that the self-management application can assist GC patients undergoing chemotherapy in managing side effects, adhering to a healthy diet and lifestyle, coordinating medication use, acquiring self-management skills, and, overall, improving their QOL [50]. Taken together, these studies illustrate the diverse strategies—novel natural products, integration of Kampo traditional medicines, and digital self-management tools—that are being explored to enhance outcomes for GC patients, highlighting an evolving therapeutic landscape.

## 4. Conclusions

Reducing the global burden of *H. pylori* infection requires integrated public health strategies that combine accurate diagnostics, regionally tailored therapies, preventive interventions, and robust surveillance. Although eradication can reduce cancer risk and improve clinical outcomes, substantial heterogeneity in prevalence, antimicrobial resistance patterns, patient adherence, and follow-up challenges complicates universal recommendations. Advances in education, including NME platforms, together with careful consideration of PPI–associated risks, are essential components of comprehensive management and stewardship. Regional treatment strategies guided by antimicrobial susceptibility testing and guideline-aligned regimens hold promise for improving eradication rates and sustaining long-term benefits. Continued research into vaccines and surveillance-enhanced program implementation will be critical to achieving durable reductions in both morbidity and cancer risk attributable to *H. pylori*. Future inquiries should address the impact of NME on *H. pylori* management and patient outcomes; regional variability and asymptomatic carriage; the trajectory of antibiotic resistance; public health implications; the efficacy and regional applicability of Vo-containing regimens for eradication; and the comparative effectiveness of quadruple, sequential, and high-dose PPI regimens, as well as the role of antimicrobial susceptibility testing in guiding therapy. In conclusion, at the individual level, successful eradication is achievable in many patients with appropriately chosen regimens, strong adherence, and post-treatment confirmation. However, there is no evidence of global eradication of *H. pylori.* The organism remains widespread, and resistance-driven regimen failure continues to challenge long-term control. Future progress hinges on personalized therapies, vaccines, and robust public health strategies that balance effective treatment with prudent antibiotic use.

In gastric and GEJ cancers, data on anti–HER-2 therapies illuminate an expanding yet heterogeneous landscape of targeted modalities, underscoring the potential of ADCs and combinatorial strategies while highlighting the ongoing need for head-to-head efficacy and safety comparisons and robust randomized trials to define optimal sequencing and patient selection. The emerging evidence on direct administration of chemotherapeutic agents in GC indicates the potential for systemic toxicity and suboptimal efficacy, emphasizing the necessity for careful monitoring, dose optimization, and protective strategies to maximize therapeutic benefit while minimizing harm. NAT incorporating ICIs shows promise for improved downstaging and resection metrics in LAGC, with tolerable safety profiles reported in several cohorts. However, definitive conclusions regarding long-term survival await prospective, randomized trials and meticulous patient selection. The elderly represent a particularly important subgroup requiring optimized regimens and vigilant safety monitoring. Additional mechanistic studies, biomarker-driven approaches, and exploration of novel combinations—such as CIK-based strategies or metronomic regimens—may further refine NAT and perioperative care in GC. The adverse effects associated with surgical and perioperative therapies reflect a landscape in which potential benefits must be weighed against heterogeneous evidence of efficacy and toxicity across modalities. High-quality, prospective research is needed to define optimal patient selection, dosing, scheduling, and supportive care strategies that maximize efficacy while minimizing toxicity in diverse surgical and oncologic settings.

The convergence of anemia, osteoporosis risk, ocular toxicity, and cancer-related fatigue highlights multifaceted challenges faced by patients undergoing chemotherapy for gastrointestinal cancers. Addressing iron deficiency anemia with agents such as ferric carboxymaltose, monitoring and mitigating osteoporosis risk, recognizing and managing rare ocular adverse events, and proactively addressing cancer-related fatigue through personalized interventions are essential components of comprehensive cancer care. A holistic approach that integrates hematologic optimization, bone health maintenance, vision monitoring, and symptom management holds promise for improving QQL and enabling safer, more effective cancer treatment. Finally, the evolving therapeutic landscape for GC encompasses diverse strategies—cellular senescence, novel natural products, integration of Kampo traditional medicines, and digital self-management tools—that are being explored to enhance patient outcomes.

Although clinicians have access to an extensive range of strategies to mitigate chemotherapy-induced toxicities, a considerable number of patients still struggle to tolerate or manage these adverse effects. To address this issue, we propose a multi-pronged strategy that emphasizes interdisciplinary collaboration, rigorous evaluation, and adaptive implementation, with the aim of translating emerging evidence into durable clinical benefits across diverse patient populations. One proposed solution to address this issue is to administer chemotherapy within internal medicine departments, in collaboration with medical oncologists and gastroenterologists, because in many cases the adverse effects outweigh the potential benefits of chemotherapy.

## Figures and Tables

**Figure 1 cancers-18-00141-f001:**
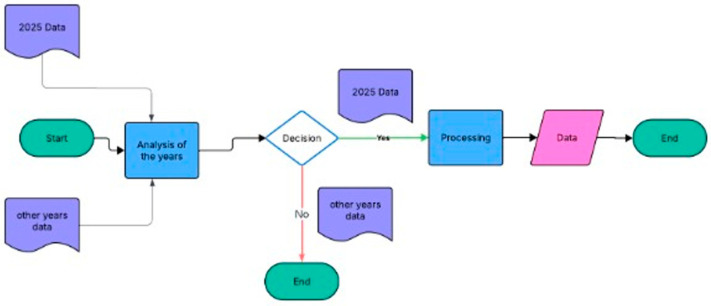
Evaluation of all data retrieval from the PubMed database.

**Figure 2 cancers-18-00141-f002:**
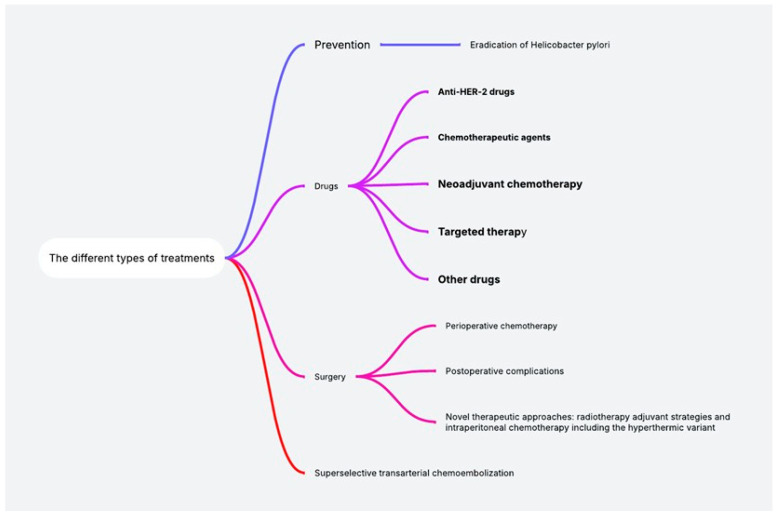
Scientific data analysis of all data retrieval from the PubMed database.

## Data Availability

Data sharing is not applicable to this article.

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
