# Peer review of "Chemotherapy for Gastric Cancer Is Not Solely the Domain of the Oncologist"

_cancers, 2025, doi:10.3390/cancers18010141_

Round 1

Reviewer 1 Report

Comments and Suggestions for Authors

Dear author

Your submitted manuscript titled "Chemotherapy for gastric cancer is not within the competence of the oncologist" is a review manuscript that emphasises chemotherapy for gastric cancer, although it is dangerous to use such chemotherapy despite other vitamin supplements helping in earlier recovery of gastric cancer. This review really helps health professionals take good care of GC patients. Overall, the manuscript is well formatted and easily understandable. Results are interesting and support the hypothesis.

I have few queries before taking a further decision.

  1. Authors should be mentioned with the time frame of data collection from the PubMed database.
  2. Establish some exclusion and inclusion criteria for a contemporary narrative review. Add a flow diagram that has all data retrieval from the PubMed database.

Author Response

Your submitted manuscript titled "Chemotherapy for gastric cancer is not within the competence of the oncologist" is a review manuscript that emphasises chemotherapy for gastric cancer, although it is dangerous to use such chemotherapy despite other vitamin supplements helping in earlier recovery of gastric cancer. This review really helps health professionals take good care of GC patients. Overall, the manuscript is well formatted and easily understandable. Results are interesting and support the hypothesis.

THANK YOU FOR YOUR KIND WORDS.

I have few queries before taking a further decision.

1.Authors should be mentioned with the time frame of data collection from the PubMed database.

THANK YOU FOR THE SUGGESTION.

IN ORDER TO EVALUATE A CLINICAL PROTOCOL, WE NEED CURRENT DATA REPORTED BY COLLEAGUES FROM OTHER COUNTRIES. FOR THIS REASON, WE ANALYZED THE DATA REPORTED IN PUBMED FOR THE YEAR 2025. WE DID NOT SELECT STUDIES BY SPECIFIC AUTHORS; WE DID NOT LOOK AT THE ORIGIN OF THE AUTHORS' WORKS. WE SELECTED 2025 STUDIES THAT WERE RELEVANT TO THE PURPOSE OF THE STUDY. THE NAMES OF THE AUTHORS OF THE SELECTED WORKS CAN BE SEEN IN THE BIBLIOGRAPHICAL REFERENCES.

WE HAVE ADDED THE FOLLOWING TO THE TEXT:

“OUR STUDY EMPLOYED A CONTEMPORARY NARRATIVE REVIEW TO IDENTIFY STUDIES PUBLISHED IN 2025 THAT REPORTED ADVERSE EVENTS ASSOCIATED WITH CHEMOTHERAPY FOR GC AND WERE AU-THORED BY CLINICAL COLLEAGUES FROM COUNTRIES OTHER THAN OUR OWN”

2.Establish some exclusion and inclusion criteria for a contemporary narrative review. Add a flow diagram that has all data retrieval from the PubMed database.

THANK YOU FOR THE SUGGESTION

WE HAVE ADDED THE FOLLOWING TO THE TEXT:

“THE INCLUSION CRITERIA ENCOMPASSED PUBLICATIONS THAT COLLECTIVELY CHARACTERIZE THE SPECTRUM OF CHEMOTHERAPY-RELATED TOXICITIES AND ADVERSE REACTIONS IN GC AND THEIR IMPLICATIONS FOR FUTURE TREATMENT STRATEGIES AND PATIENT QQL. ELIGIBLE STUDIES WERE SCREENED FOR RELEVANCE, AND THE FINDINGS WERE SYNTHESIZED TO MAP THE CURRENT KNOWLEDGE BASE AND IDENTIFY GAPS IN THE LITERATURE. EXCLUSION CRITERIA WERE APPLIED TO ALL YEARS EXCEPT 2025.”

WE HAVE INTRODUCED THE A FLOW DIAPHRAGM IN THE TEXT (SEE FIGURE 1 IN TEXT)

Reviewer 2 Report

Comments and Suggestions for Authors

This article, "Chemotherapy for Gastric Cancer is Not Within the Competence of Oncologists", provides a certain amount of review and description, but there are significant problems with this article. Firstly, the title and the theme do not match. The description of the key innovative aspects of this research is not prominent, and the clinical guidance significance is not highlighted either. At the same time, the issues raised in this article have not been resolved, and the enlightening significance of this article is very limited. Regarding the description of the current state of gastric cancer research, it lacks innovation and is rather superficial. I believe the author should clarify the significance of this article and its overall structure.

Author Response

This article, "Chemotherapy for Gastric Cancer is Not Within the Competence of Oncologists", provides a certain amount of review and description, but there are significant problems with this article. Firstly, the title and the theme do not match.

THANK YOU FOR THE SUGGESTION. 

  1. WE HAVE EXPLAINED THE TOPIC AND OBJECTIVES OF THE ARTICLE BETTER IN THE TEXT. SEE IN THE TEXT:

“THEME AND OBJECTIVE

THIS MANUSCRIPT DRAWS ITS THEME FROM ROUTINE CLINICAL PRACTICE. IN GC, A SUBSTANTIAL PROPORTION OF PATIENTS ARE DISCHARGED HOME BETWEEN CHEMOTHERAPY CYCLES TO MANAGE TREATMENT-RELATED ADVERSE EFFECTS INDEPENDENTLY; IN MANY CASES, THE MEDICATIONS PRE-SCRIBED TO MITIGATE THESE TOXICITIES FAIL TO PROVIDE ADEQUATE RELIEF. THESE ADVERSE EFFECTS ARE WELL DOCUMENTED AND AFFECT AN INCREASING NUMBER OF PATIENTS. THE STUDY DOES NOT AIM TO IDENTIFY NOVEL ASPECTS OF CARE. RATHER, ITS PRIMARY OBJECTIVE IS TO EVALUATE THE ADVERSE EFFECTS ASSOCIATED WITH CHEMOTHERAPY IN PATIENTS WITH GC. A SECONDARY OBJECTIVE IS TO SUMMARIZE CURRENT RESEARCH IN THE FIELD TO CONTEXTUALIZE THE ISSUE; HOWEVER, THIS WORK IS NOT INTENDED TO BE A COMPREHENSIVE REVIEW. INSTEAD, THE SCOPE IS RESTRICTED TO EXAMINING INTERVENTIONS THAT REDUCE CHEMOTHERAPY-ASSOCIATED TOXICITY IN GC PATIENTS.”

2.REGARDING THE TITLE, WE FOLLOWED THE SUGGESTION OF REVIEWER 3 AND CHANGED THE TITLE TO:

“CHEMOTHERAPY FOR GASTRIC CANCER IS NOT SOLELY THE DOMAIN OF THE ONCOLOGIST”

The description of the key innovative aspects of this research is not prominent, and the clinical guidance significance is not highlighted either.

THANK YOU FOR THE SUGGESTION. 

WE HAVE EXPLAINED THIS ASPECT BETTER IN THE TEXT. SEE IN THE TEXT:

“A SECONDARY OBJECTIVE IS TO SUMMARIZE CURRENT RESEARCH IN THE FIELD TO CONTEXTUALIZE THE ISSUE; HOWEVER, THIS WORK IS NOT INTENDED TO BE A COMPREHENSIVE REVIEW. INSTEAD, THE SCOPE IS RESTRICTED TO EXAMINING INTERVENTIONS THAT REDUCE CHEMOTHERAPY-ASSOCIATED TOXICITY IN GC PATIENTS.”

At the same time, the issues raised in this article have not been resolved, and the enlightening significance of this article is very limited.

THANK YOU FOR THE SUGGESTION. 

WE RAISED ONLY 1 ISSUE:

DESPITE THE RANGE OF MEDICATIONS PRESCRIBED BY ONCOLOGISTS TO MANAGE CHEMOTHERAPY-INDUCED TOXICITIES, MANY PATIENTS CONTINUE TO STRUGGLE TO COPE WITH THEM

WE PROPOSED:

SEE IN ABSTRACT:

WE RECOMMEND THAT THESE CHEMOTHERAPY REGIMENS BE ADMINISTERED WITHIN INTERNAL MEDICINE DEPARTMENTS, IN COLLABORATION WITH THE MEDICAL ONCOLOGIST AND GASTROENTEROLOGIST, BECAUSE IN MANY CASES THE ADVERSE EFFECTS OUTWEIGH THE POTENTIAL BENEFITS OF CHEMOTHERAPY.

WE HAVE EXPLAINED THIS ASPECT BETTER IN CONCLUSIONS. SEE IN THE TEXT:

“ALTHOUGH CLINICIANS HAVE ACCESS TO AN EXTENSIVE RANGE OF STRATEGIES TO MITIGATE CHEMOTHERAPY-INDUCED TOXICITIES, A CONSIDERABLE NUMBER OF PATIENTS STILL STRUGGLE TO TOLERATE OR MANAGE THESE ADVERSE EFFECTS. TO ADDRESS THIS ISSUE, WE PROPOSE A MULTI-PRONGED STRATEGY THAT EMPHASIZES INTERDISCIPLINARY COLLABORATION, RIGOROUS EVALUATION, AND ADAPTIVE IMPLEMENTATION, WITH THE AIM OF TRANSLATING EMERGING EVIDENCE INTO DURABLE CLINICAL BENEFITS ACROSS DIVERSE PATIENT POPULATIONS. ONE PROPOSED SOLUTION TO ADDRESS THIS ISSUE IS TO ADMINISTER CHEMOTHERAPY WITHIN INTERNAL MEDICINE DEPARTMENTS, IN COLLABORATION WITH MEDICAL ONCOLOGISTS AND GASTROENTEROLOGISTS, BECAUSE IN MANY CASES THE ADVERSE EFFECTS OUTWEIGH THE POTENTIAL BENEFITS OF CHEMOTHERAPY.”

Regarding the description of the current state of gastric cancer research, it lacks innovation and is rather superficial. I believe the author should clarify the significance of this article and its overall structure.

THANK YOU FOR THE SUGGESTION. 

WE HAVE EXPLAINED THIS ASPECT BETTER IN CONCLUSIONS. SEE IN THE TEXT:

“A SECONDARY OBJECTIVE IS TO SUMMARIZE CURRENT RESEARCH IN THE FIELD TO CONTEXTUALIZE THE ISSUE; HOWEVER, THIS WORK IS NOT INTENDED TO BE A COMPREHENSIVE REVIEW. INSTEAD, THE SCOPE IS RESTRICTED TO EXAMINING INTERVENTIONS THAT REDUCE CHEMOTHERAPY-ASSOCIATED TOXICITY IN GC PATIENTS.”

Reviewer 3 Report

Comments and Suggestions for Authors

The referenced article is a narrative review of gastric cancer treatments. It proposes that chemotherapy treatments be administered in the internal medicine service due to their wide and diverse range of side effects.  

The topic is highly relevant and of interest in the field, as it delves deeply into the most recent therapies. The methodology is sound; since it is not a systematic review, it is not mandatory to list all the sources and selection criteria used. However, it would be advisable to explain which sources (databases) were used for the review, as well as how the included references were chosen. The conclusions are consistent with the evidence and arguments and address the research objectives.  

I believe it is an excellent and thorough literature review. Most of the references are from this year. It is completely up-to-date.

The title seems very challenging to me; perhaps it should say that it is not solely the domain of the oncologist.  

The authors are invited to create a visual diagram of the different types of treatments and the lines used.

Author Response

The referenced article is a narrative review of gastric cancer treatments. It proposes that chemotherapy treatments be administered in the internal medicine service due to their wide and diverse range of side effects.  

The topic is highly relevant and of interest in the field, as it delves deeply into the most recent therapies. The methodology is sound; since it is not a systematic review, it is not mandatory to list all the sources and selection criteria used. However, it would be advisable to explain which sources (databases) were used for the review, as well as how the included references were chosen. The conclusions are consistent with the evidence and arguments and address the research objectives.  

THANK YOU FOR YOUR KIND WORDS. REGARDING THE METHODOLOGY, WE HAVE INTRODUCED THE FOLLOWING WORDS AND FIGURE 1 INTO THE TEXT.

“METHODOLOGY

OUR STUDY EMPLOYED A CONTEMPORARY NARRATIVE REVIEW TO IDENTIFY STUDIES PUBLISHED IN 2025 THAT REPORTED ADVERSE EVENTS ASSOCIATED WITH CHEMOTHERAPY FOR GC AND WERE AUTHORED BY CLINICAL COLLEAGUES FROM COUNTRIES OTHER THAN OUR OWN. PUBMED WAS SEARCHED USING THE TERM “SIDE EFFECTS OF CHEMOTHERAPY IN GASTRIC CANCER” TO IDENTIFY THE MOST SIGNIFICANT AND RECENT PUBLICATIONS ON THE TOPIC. THE INCLUSION CRITERIA ENCOMPASSED PUBLICATIONS THAT COLLECTIVELY CHARACTERIZE THE SPECTRUM OF CHEMOTHERAPY-RELATED TOXICITIES AND ADVERSE REACTIONS IN GC AND THEIR IMPLICATIONS FOR FUTURE TREATMENT STRATEGIES AND PATIENT QQL. ELIGIBLE STUDIES WERE SCREENED FOR RELEVANCE, AND THE FINDINGS WERE SYNTHESIZED TO MAP THE CURRENT KNOWLEDGE BASE AND IDENTIFY GAPS IN THE LITERATURE. EXCLUSION CRITERIA WERE APPLIED TO ALL YEARS EXCEPT 2025.”

I believe it is an excellent and thorough literature review. Most of the references are from this year. It is completely up-to-date.

THANK YOU FOR YOUR KIND WORDS.

The title seems very challenging to me; perhaps it should say that it is not solely the domain of the oncologist.  

WE AGREE. AS SUCH, WE HAVE CHANGED THE TITLE IN ACCORDANCE WITH YOUR SUGGESTION.

TITLE:

“CHEMOTHERAPY FOR GASTRIC CANCER IS NOT SOLELY THE DOMAIN OF THE ONCOLOGIST”

The authors are invited to create a visual diagram of the different types of treatments and the lines used.

THANK YOU FOR THE SUGGESTION. WE HAVE INTRODUCED FIGURE 2. SEE IN TEXT.